# An Image Detection Method for Image Stabilization Deviation of the Tank Gunner’s Primary Sight

**DOI:** 10.3390/s23115039

**Published:** 2023-05-24

**Authors:** Zhannan Guo, Baoqi Xie, Yingshun Li, Ximing Sun

**Affiliations:** 1School of Control Science and Engineering, Dalian University of Technology, Dalian 116024, China; guozhannan@mail.dlut.edu.cn (Z.G.);; 2School of Automation, Guangxi University of Science and Technology, Liuzhou 545000, China

**Keywords:** the gunner’s primary sight control system, image stabilization deviation, YOLOv5, *δ-SIOU*, the spatial pyramid pool module, BiFPN network, C3CA

## Abstract

The primary sight control system of a tank gunner has image stabilization as one of its primary functions. The image stabilization deviation in the aiming line is a key indicator for evaluating the operational status of Gunner’s Primary Sight control system. Employing image detection technology to measure image stabilization deviation enhances the effectiveness and accuracy of the detection process and allows for the evaluation of image stabilization functionality. Hence, this paper proposes an image detection method aimed at the Gunner’s Primary Sight control system of a specific tank which utilizes an enhanced You Only Look Once version 5 (YOLOv5) sight-stabilizing deviation algorithm. At first, a dynamic weight factor is integrated into SCYLLA-IoU (*SIOU*), creating *δ-SIOU*, which replaces Complete IoU (*CIoU*) as the loss function of YOLOv5. After that, the Spatial Pyramid Pool module of YOLOv5 was enhanced to improve the multi-scale feature fusion ability of the model, thereby elevating the performance of the detection model. Finally, the C3CA module was created by embedding the Coordinate Attention (CA) attention mechanism into the CSK-MOD-C3 (C3) module. The Bi-directional Feature Pyramid (BiFPN) network structure was also incorporated into the Neck network of YOLOv5 to improve the model’s ability to learn target location information and image detection accuracy. Based on data collected by a mirror control test platform, experimental results indicate an improvement in the detection accuracy of the model by 2.1%. These findings offer valuable insights into measuring the image stabilization deviation in the aiming line and facilitating the development of the parameter measurement system for Gunner’s Primary Sight control system.

## 1. Introduction

The Gunner’s Primary Sight control system is a critical component of modern tank fire control systems that ensures accurate target distance measurement and aiming in all weather conditions. The image stabilization scope is among the principal features of the aiming scope, enabling the gunner to observe the target and background steadily even while the tank is in motion [1]. A gyroscope is employed to stabilize the mirror in the optical path of the scope, compensating for any sight-line deviation caused by the vehicle’s jerking motion [2]. By measuring image stabilization deviation, the functionality of the image stabilization scope can be appropriately ensured.

The traditional image detection methods utilize manual feature extraction techniques such as the Sobel edge detection feature [3], Haar feature [4], and Hog feature [5], among others. These methods primarily depend on features that were designed manually and require professionals with specialized knowledge and a complex parameter adjustment process, resulting in limited generalization ability and robustness. In recent times, with the emergence of deep learning, image detection is progressively transitioning into this deep learning era [6]. It is effectively applied in multiple fields, such as carrier chip defect detection [7], remote sensing image detection [8], plant disease recognition [9], and scratch recognition and positioning [10], among other areas. Deep learning models possess strong fitting and reasoning capabilities while also providing robust and generalizable abstract feature extraction. Some scholars use the deep learning model; achieving correct identification and detection of crop diseases may save the crops from damage [8]. In [11], the author introduces a one-time aggregation module to optimize the backbone network structure under the YOLOv5s network model framework, and experiments are conducted on a self-made vehicle detection dataset. The experiment shows that the average accuracy of different object detections is improved, and the detection speed meets the real-time requirements. Envelope W. et al [12]. employed an improved version of the Siamese network architecture and incorporated *DIoU* as a novel loss function, reducing the influence of interference factors in the complex background on the tracker. In addition, deep learning has also been applied in the field of medicine. Research [13] proposes a new image segmentation method based on the K-means clustering algorithm (KMC) and novel fast-forwards quantum optimization algorithm (FFQOA), accomplishing a new early screening method for the investigation of COVID-19 pneumonia using chest CT scan images. To eliminate subjectivity and limitations inherent in traditional deviation detection, a deep neural network was implemented to automatically detect the aiming line in image stabilization images and calculate deviations. This approach enhances efficiency and accuracy [14].

At present, deep neural networks are commonly used in target detection algorithms, which can be divided into two categories. The first type is a one-stage detection algorithm based on YOLO or SSD [15]. The other type is a two-stage detection algorithm based on R-CNN, Faster R-CNN [16], and other methods. Unlike one-stage detection methods that directly regress borders, in the two-stage detection algorithm, a sequence of candidate regions is generated by convolution. Additionally, other algorithms for target classification are employed to improve the accuracy of object detection.

Although the two-stage algorithm is highly accurate but increases the test time extensively [17], in practical engineering, real-time detection is often required. By enhancing the network structure and incorporating attention mechanisms, the one-stage detection algorithm can be improved, ensuring both real-time performance and accurate image detection [18]. This study adopts a faster one-stage detection algorithm that detects the image stabilization of a gunner’s sight. This paper proposes an improved YOLOv5 model for detecting image stabilization errors in a gunner’s sight. The model builds upon YOLOv5 to enhance its detection performance. The main research content is as follows:In this paper, we employed the image collection device of Gunner’s Primary Sight control system to collect the image data of the aiming line in the scope of sight. We created a dataset for the experiment and enhanced it to prevent overfitting issues and improve the accuracy of the model detection;To enhance the performance of YOLOv5’s backbone network, the SPPF module should be replaced with the SPPCSPC module. This module improves the model’s multi-scale target fusion ability, detection speed, and detection performance;Introduce the Coordinate Attention (CA) in YOLOv5 to enhance the attention of the model of the aiming line. Moreover, embed the CA into the C3 module to form a new C3CA module;In this paper, we propose a solution for neck network improvement by combining the BIFPN and C3CA modules. Our solution enhances the model’s characteristics across different scales resulting in improved semantic as well as location information which ultimately leads to better detection accuracy;The paper utilized the *SIoU* loss function with dynamic weights and improved it further, resulting in the *δ-SIOU* loss function. The experiments conducted proved that the proposed model not only achieved a high detection accuracy but also had a fast detection speed.

The rest of this paper is organized as follows. In Section 2, we introduce the image acquisition device of Gunner’s Primary Sight and establish the aiming line data set used in the experiment. Section 3 describes our research on aiming line detection. Section 4 presents our experimental results and analysis. Section 5 summarizes our work and provides some suggestions for future work.

## 2. Data Preparation

### 2.1. The Collection Device

At present, most of the world’s advanced tanks adopt a high-precision image-stabilized fire control system. The image-stabilized fire control system allows the gunner to provide a relatively stable observing environment when observing the target and background using the sight mirror. The stabilized fire control system mainly controls image stabilization through upper or lower reflection. The up-reflection image-stabilized fire control system relies on the gyro stabilization platform as its core. The light path structure of the upper reflection image-stabilized fire control system is simple and only stabilizes the upper reflector sight. It provides a large and stable vision. Stable field of view, aiming view will not turn black when the tank is driving on uneven roads or crossing craters. The upper reflector image-stabilized fire control system can still ensure stable image stabilization under the work of night vision and thermal image. Therefore, most newly developed fire control systems worldwide adopt the upper reflector image-stabilized control.

This thesis focuses on examining the up-reflection of Gunner’s Primary Sight control system. A mirror control test platform connects the primary components of Gunner’s Primary Sight control system, and the system is fully constructed using dynamic simulation technology. An upper computer controls the test platform and simulates the steady-image condition of the tank’s working status. Use the collection device displayed in Figure 1 to gather the image-stabilized sight image.

The collimator is primarily composed of a mirror body and an optical system. The lighting group is installed in front of the matte glass collimator, with a rating adjustment range from 0 to 500 Lx. A lighting group with an adjustable brightness range from 0 to 500 Lx is installed in front of frosted glass. The installation bracket frame of the collimator is constructed from stainless steel and has an adjustable thickness of the steel plate. Figure 2 depicts the schematic diagram of the collimator installation. The reticle has small squares, with each measuring at an angle value of 4’, amounting to 60 small grids per line, and with a total of 60 lines.

### 2.2. Aiming Line Image Example

The image collection process in the CCD system employs industrial cameras with WP-UT2000M from WORK POWER. Equipped with the USB 3.0 interface, the camera boasts a remarkable capacity for collecting and transmitting data quickly. Moreover, the megapixel capability is as high as 20, and the frame rate is 18 per second. Mounting on the mirror control system bracket, the Gunner’s Primary Sight has both the camera and the stabilizing head and the lower mirror body fixed on it. Imitating the gunner’s sight system function in a specific battle circumstance is facilitated via the control device. The camera consistently gathers images under varying light and position. Figure 3 illustrates partial data on the accumulation of aiming lines.

The images of the aiming line were captured several times using a camera placed in different positions and lighting angles by a collection device. Thirteen videos were recorded and collected, which were then segmented into individual frames. Similar frames from the same video were screened and sampled, and 8629 images were collected to create a dataset of aiming line images used in this study. The aiming line images were labeled using Labelme software. The dataset was divided into training, test, and verification sets in a ratio of 7:2:1, as shown in Table 1.

In this paper, we aim to enhance the accuracy of the detection model and prevent overfitting using various techniques such as random rotation, translation, scaling, and cutting [19]. As a result, a dataset will be built and utilized in the experiment reported in this paper. Part of the effect diagram after the data enhancement is shown in Figure 4.

## 3. Improve YOLOv5 Model

In this section, an Introduction to the YOLOv5 network is first presented. Next, based on the basic YOLOv5 structure, replace the SPPF with the SPPCSPC module, and embed the CA into the C3 module to form a new C3CA module. In addition, the combination of BIFPN structure is employed to modify the YOLOv5s structure, to improve the accuracy of aiming line detection are introduced. In the end, the improved *SIoU* (*δ-SIOU*) is used as the loss function of our model. Figure 5 shows the diagram of the process for applying the modified YOLOv5s model to aiming line detection of Gunner’s Primary Sight.

### 3.1. YOLOv5 Basic Model

The YOLO series is a classic and widely-used algorithm within the one-stage detection approach. Several members of the YOLO series have been proposed over time, including YOLOV1 [20], Yolov2 [21], YOLOV3 [22], YOLOV4 [23], YOLOV5 [8], and many others. Its target detection performance has been continuously improved. In this paper, we adopt the smaller-sized and more accurate YOLOV5 model as our neural network structure. As part of the YOLO family, YOLOv5 is a representative end-to-end target detection network. The YOLOV5 network structure consists of four main components: input layer, backbone network, neck network, and head network. Figure 6 presents the structure of the YOLOV5 network.

As shown in Figure 6, the detected image is fed into the network’s input end. The backbone network then extracts target features in the image while the neck network integrates multi-scaled features. Lastly, based on features extracted by the neck network, the head network generates the feature vector utilized in generating the boundary box and predicting the categories in the image.

### 3.2. Improved Spatial Pyramid Pooling—SPPCSPC

Since its launch in 2021, YOLOv5 has undergone updates, specifically in version YOLOv5_6.0, where the original SPP module was replaced with the SPP structure, as illustrated in Figure 7, thus increasing the rate under the same conditions. To enhance the model’s detection performance, this study adopts a new spatial pyramid pooling module, specifically the SPPCSPC structure. This structure employs three different sizes of pooling cores for better detection. This structure employs three different sizes of pooling cores. To further integrate the detection target characteristics, the study introduced a novel pooling method based on a nuclear structure and parallel convolution nuclear structure for better detection. Figure 7 presents the SPPCSPC structure. The literature [24] reports the superiority of the SPPCSPC structure over SPP and SPPF in terms of improving the model’s ability to fuse multi-scale targets. Figure 7 illustrates the SPPCSPC structure.

The actual application of the Space Pyramid Pool Compressible Split-Attention Convolution (SPPCSPC) module exhibits better performance than SPP and SPPF structures. The SPPCSPC structure improved the model’s ability to fuse targets across multiple scales. This study involved the generation of a group of 32,512,256,256 random numbers, which were iteratively processed through the three structures for a total of 50 experimental rounds. The comparative results of the experimental operation speed, based on the requirements of the three structures, are presented in Table 2. The speed comparison table shows that the processing speed of random numbers using the SPPCSPC is faster than the processing speed of the other two structures.

### 3.3. Attention Mechanism

#### 3.3.1. Coordinate Attention Mechanism Structure

The attention mechanism has been a significant development in the image and natural language processing and has demonstrated its efficacy in enhancing model performance [25]. The parameters of deep learning models, which increase to improve their expressiveness during neural network learning, often result in information overload. By focusing on the most relevant information amidst the available information, the attention mechanism enables models to reduce attention on extraneous information, thereby improving task efficiency and accuracy [26]. It can reduce the attention of irrelevant information to improve the efficiency and accuracy of task processing of the model.

This paper aims to detect the aiming line error in the image stabilization of a specific tank Gunner’s Primary Sight control system under steady-image conditions. To improve the detection capacity of the model for the aiming line, this paper introduces the coordinate attention mechanism (CA). By enhancing the model’s attention to the location information of the aiming line, this study improves its accuracy in recognizing the aiming line. Figure 8 shows the structure of the CA attention module.

The CA attention mechanism incorporates information about the target location into the channel [14]. The CA attention mechanism replaces the original two-dimensional global pooling with two one-dimensional pools for feature coding. The CA can aggregate along two directions, capture remote dependencies in one, and retain precise location information in the other. Finally, the generated feature maps are encoded to obtain directional and locational information. Although the global pooling method is commonly used to encode global spatial information, it compresses this information and makes retaining location details difficult. To better capture accurate location information, the CA transformed global pooling into a one-dimensional feature encoding operation.

First, select two pooling cores with dimensions (*H*, 1) and (1, *W*), respectively, which will be applied along the horizontal and vertical directions of each channel. The expression for the output of a specific channel with height h can be represented as follows:(1)$$z_{c}^{h}=\frac{1}{W}\sum_{0\leq i\leq W}x_{c}(h,i)$$

In the same way, the output of a c-channel with a width of w can be represented as:(2)$$z_{c}^{w}=\frac{1}{H}\sum_{0\leq j\leq W}x_{c}(j,w)$$
where $x_{c}$ is the eigentensors.

Through the above two kinds of transformation, feature aggregation is carried out independently for the two spatial directions, generating a pair of directional perception feature maps. This method allows the channel attention to capture long-range dependencies in one direction while preserving accurate location information in the other direction. This improves the model’s positional accuracy.

#### 3.3.2. Extended C3 Module

The C3 module is a prevalent component of the YOLOV5 model, which forms the backbone and neck networks. It is a network model that incorporates the CSPNet network structure design concept into the fundamental residual model. Structurally, it enhances the CNN learning capability, minimizes internal loss, and decreases calculation bottlenecks. By incorporating the C3 module, it can cut down on the computational costs of network models while also ensuring accuracy, leading to more optimal industrial use. The specifics of the C3 module’s structure are shown in Figure 9.

The C3 module uses two parallel convolution operations to reduce the size of the feature maps by half. One of the branches links to the Bottleneck module after the Conv layer, while the feature diagram of the other branch connects through the Concat operation. The concatenated feature maps undergo C3 module processing and output the final feature map after the convolutional layer.

In this paper, we have implemented the Channel Attention (CA) mechanism within the C3 module. We have connected the CA module to the C3 module that lacks the Bottleneck’s support. The input to the CA attention module is given through a convolutional layer. The two parallel branches are then operated and connected through the Concat layer. The remaining part of the C3 module is kept unchanged. As shown in Figure 10, this integration has led to the creation of the C3CA module.

### 3.4. Improved Neck Network

Unlike the top-down characteristic fusion network, YOLOV3, which is depicted in Figure 11a, YOLOV5 introduced the bottom-up characteristic pyramid structure based on the original FPN feature fusion network. This structure, called FPN + PAN, is depicted in Figure 11b. The use of this structure enhances both semantic and location information, leading to improved detection accuracy of the model’s object targets [25]. However, FPN + PAN only merges features of the same size during feature merging, resulting in inadequate fusion of the characteristics for different scales of the model. To address this, we propose the bi-directional Feature Pyramid Network structure (BIFPN), which is depicted in Figure 11c. BIFPN more fully fuses different scale features, further improving the model’s ability to detect objects.

The BiFPN incorporates cross-scale connections and weighted characteristic fusion to reduce the scales and increase efficiency. First, to simplify the structure, it removes nodes that contribute insignificantly to the feature network. These are nodes that have a solitary input and are not merged with other inputs.Cross-scale connection: Initially, the structure is simplified by removing nodes that do not contribute significantly to the feature network. This involves having only one node that does not fuse with any other input edges. Further, to attain a higher level of feature fusion, each bidirectional path is considered a feature network for repeated superposition;Weighted Feature Fusion: Conventional methods may result in the loss of effective information due to the addition of features after unifying scales of varying features. To address this, we have adopted a Fast Normalized Fusion (FNF) feature based on the softmax function. Formula (3) shows that the FNF feature effectively retains more information during the characteristic fusion process while also accelerating the training speed;

(3)$$O=\sum_{i}\frac{w_{i}}{\varepsilon +\sum_{j}w_{j}}\cdot I_{i}$$where *O* is the output. *I* is input. $w_{i}$ is a learning weight. ε is learning rate, which is usually 0.0001 to avoid numerical instabilities.

The BiFPN was developed by integrating bidirectional cross-scale connections and fast normalized fusion. To illustrate this point, consider layer P3 in Figure 12. The fusion characteristics are calculated using the following formula:(4)P3td=Convw1×P3+w2×Resize(P4in)w1+w2+ε
(5)P3out=Convw3×P3+w4×P3+w5×Resize(P2out)w3+w4+w5+ε

The convolution operation is denoted by *Conv*, while the operations for upsampling or downsampling are represented by Resize. Figure 12 illustrates the remaining parameters and operations.

This paper introduces the above-mentioned structure into YOLOv5 to improve its ability to fuse multi-scale features. The C3CA module, which incorporates the CA attention mechanism, was created by embedding it into the C3 module. Only the C3 modules of the original neck network were replaced to achieve accurate and efficient model detection. This decision was made based on the following reasons:3.If all C3 modules in the YOLOv5 network are substituted with C3CA modules, it will introduce a substantial number of training parameters. This would lead to increased training time and decreased detection speed due to the complex network structure;4.Our findings suggest that relying solely on the C3CA module results in a suboptimal detection accuracy for network structures that lack the C3 module.

The improved network structure in YOLOv5 is shown in Figure 13.

### 3.5. Improved Loss Function—δ-SIOU

At first, the majority of target detection algorithms use Intersection over Union (*IoU*) to gauge the precision of identifying the relevant objects during training. The absence of distance measurement between the real box and the predicted box when using *IoU* as a loss function results in certain issues. Firstly, non-intersecting boxes fail to capture the magnitude of the span and percentage of overlap, which leads to a loss of zero, no gradient return, and the inability to train the model. In addition, it is impossible to differentiate between various alignment approaches for the two objects. Specifically, when two overlapping objects have similar differences in varying angles, their corresponding *IoU* values will be identical.

In view of the above problems, continuous improvements have been applied to target detection, mainly *GIOU* [27], *DIOU* [28], *CIOU* [24], *EIOU* [29], and *SIOU* [30]. Due to given space constraints, a detailed explanation of each will be omitted. To determine the optimal loss function, we utilized the image-stabilization dataset for Gunner’s Primary Sight. Using the YOLOv5 network model, we conducted a training experiment with 200 rounds of comparison, resulting in the regression loss curve represented in Figure 14.

As can be seen from Figure 15, the *SIOU* outperformed other loss functions in terms of both convergence speed and final loss size on the dataset used in this study. Therefore, this research adopts the *SIOU* as the loss function and enhances it.

The *SIOU*, as shown in Figure 15, redefines the correlation-loss function to accelerate the convergence rate by introducing the vector angle between the real and predicted boxes. The *SIOU* includes the following four parts: determining the intersection, computing the union, calculating the vectors’ angle, and using the angle in the loss function.5.The angle loss Λ:
(6)$$\Lambda =1-2*\sin^{2}(\arcsin(\frac{c_{h1}}{\sigma })-\frac{\pi }{4}))$$where the $c_{h1}$ is the height difference between the center point of the real box *A* and the predicted box *B*. The σ is the distance between the center point of the real box *A* and the prediction box *B*.6.The distance loss Δ:(7)$$\left\{\begin{array}{c} \Delta =2-e^{-\gamma x}-e^{-\gamma y}\\ x={\left(\frac{x_{B}-x_{A}}{c_{w2}}\right)}^{2},y={\left(\frac{y_{B}-y_{A}}{c_{h2}}\right)}^{2},\gamma =2-\Lambda \end{array}\right.$$
where $c_{w2},c_{h2}$ respectively means the width and height of the minimum external rectangle of the real box *A* and the predicted box *B*. $\left(x_{A},y_{A}),(x_{B},y_{B}\right)$ respectively means the center coordinates of the real box *A* and the predicted box *B.*7.The shape loss Ω:(8)$$\left\{\begin{array}{l} \Omega ={\left(1-e^{w}\right)}^{\theta }+{\left(1-e^{h}\right)}^{\theta }\\ w=\frac{\left|w_{A}-w_{B}\right|}{\max(w_{A},w_{B})},h=\frac{\left|h_{A}-h_{B}\right|}{\max(h_{A},h_{B})} \end{array}\right.$$
where $\left(w_{A},h_{A}),(w_{B},h_{B}\right)$ are the width and height of the real box *A* and the predicted box *B*. The θ usually within [2,6].8.The final loss:(9)$$L_{SIoU}=1-IoU+\frac{\Delta +\Omega }{2}$$

However, the *SIoU* calculates the distance and shape loss with the same weight, leading to equal influence on various constraints during training. When the distance loss is smaller, the shape loss becomes more significant. To address this issue, a dynamic weight *δ* is introduced in this study to modify the impact of different constraints on individual boxes. Specifically, the weight *δ* is determined by the inverse square root of the area of the predicted box. The specifics are as follows:(10)$$L_{SIoU}=1-IoU+\delta *\Delta +(1-\delta )*\Omega$$

According to the above analysis, the study found a negative correlation between *IoU* and distance loss but a positive correlation between *IoU* and shape loss. Given that *IoU* values range between 0 and 1 [11], this paper employs a dynamic regulating factor, represented as $\delta =e^{\left(-nIoU\right)}$ (where *n* is a constant), to modify the impact of different constraints. Thus, the previous datasets were trained under similar conditions to this study, and the outcomes are displayed in Figure 16. The enhanced *SIoU* shows superior convergence performance and fewer loss errors.

## 4. Experiment Analysis

### 4.1. Experimental Configuration and Parameter Settings

In this paper, the Pytorch 1.10 framework is adopted, and Python3.6 is used for experimental verification under the Windows 11 operating system. Other configurations are shown in Table 3.

The pictures used in the model training process are in 2740 × 1824 size JPG format. The learning rate is set to 0.001. The batch size is set to 16. The IoU threshold is set to 0.8. The number of iterations is set to 200 Epochs.

### 4.2. Evaluation Indication

To evaluate the accuracy and effectiveness of the model, the performance is measured using the mean average precision (*mAP*) [31]. The calculation of the model’s *mAP* involves computing the recall (*R*) [32], precision (*P*) [33], and the average accuracy for a particular class (*AP*). The calculation formula for precision and recall is shown below.
(11)$$\left\{\begin{array}{c} P=\frac{TP}{TP+FP}\\ R=\frac{TP}{TP+FN} \end{array}\right.$$
where the *TP* represents the count of positive samples that are correctly identified, whereas the *FP* is the count of negative samples that are wrongly identified as positive; conversely, the *FN* represents the count of positive samples that are wrongly identified as negative. This terminology is commonly used in classification models in statistical analyses.

The *AP* represents the area under the Precision-Recall curve with Precision (*P*) as the vertical axis and Recall (*R*) as the horizontal axis [31]. The *mAP*, on the other hand, refers to the average Precision values for a given category. The formulas used for their calculation are as follows:(12)$$AP=\int_{0}^{1}P(R)dR$$
(13)$$mAP=\frac{1}{n}\sum_{i=1}^{n}AP_{i}$$

### 4.3. Analysis of Experimental Results

Regarding the improvements *SIoU* adopted in this paper, the coefficients were selected for training using the above-mentioned dataset, and the specific loss curves under different coefficients are shown in Figure 17. From the comparison of the loss curve graph, it can be seen that when the coefficient is greater than 5, the initial value of the loss curve is lower, and the final loss is also smaller, indicating that the convergence of regression loss is better when the coefficient is greater than 5 compared to other coefficients. When the coefficient is 6 and 7, the overlap degree of the loss curves is high, and the difference in the final loss is also small. To further determine the coefficient in the selection, the average accuracy was calculated when the coefficient was 6 and 7, as shown in Table 4. When the coefficient was 6, the accuracy was 94.8%, while when the coefficient was 7, it decreased by 0.2 percentage points, indicating that there was no significant difference in the final loss error of the target box when the coefficient was greater than 6, but it was lower than when the coefficient was 6. Therefore, when training the improved YOLOv5 aiming line detection model in this paper, the adjustment factor δ used is *δ-SIOU* adopted $\delta =e^{\left(-6IoU\right)}$.

According to the experimental results, this paper is selected $\delta =e^{\left(-6*IoU\right)}$ as a dynamic adjustment factor for *δ-SIOU*. To verify the performance of the model proposed in this study in detecting the aiming line of Gunner’s Primary Sight, we conducted training using the dataset established in this paper. Using the dataset created in this paper, we conducted training for the detection of the aiming line of the gunner’s primary sight. The training results are shown in Figure 18. From Figure 18a, the loss convergence rate is found to be greater, and the final loss is lesser in the training process when compared to YOLOv5. Furthermore, the precision, recall, and *mAP* of the model proposed in this study are higher than those of YOLOv5. This proves that the model proposed in this study is more accurate than YOLOv5. The analysis of the training results reveals that the method proposed in this study is more efficient and accurate than YOLOv5 in detecting the aiming line of Gunner’s Primary Sight.

#### 4.3.1. Ablation Experiment

The paper aimed to determine the influence of each improved module in the improved YOLOv5 on the performance of aiming line detection. Moreover, to provide stronger evidence of the proposed method’s advantages, an ablation experiment was conducted. In this experiment, 15 groups of experiments were randomly combined by pairing each improvement module and performed on the same aiming line dataset. The results of the ablation experiments are summarized in Table 5.

According to the ablation experiment, the single use of the *δ-SIOU* can improve the detection performance to some extent, while the other single module has no significant improvement in the detection performance of the improved basic model. Moreover, using the BIFPN structure alone affects the detection performance of the model. However, the combination of the C3CA module and BiFPN structure can significantly improve the detection performance of the aiming line. Through analysis, the introduction of BiFPN structure, while improving the model’s characteristics fusion capabilities, makes the structure of the model more complicated and reduces the learning ability of the model. After the C3CA module is embedded, the C3CA module can enhance the feature information extraction ability of the model for the target. Therefore, the combination of the BiFPN structure and C3CA module significantly improves the model detection performance. Compared with other improved modules, *δ-SIOU* can improve the detection performance when used singularly or in combination, indicating that the model detection performance improvement proposed in this paper offers obvious advantages. For aiming line detection, the improved YOLOv5 model significantly improved the detection performance. It shows that the model designed in this paper gives full play to the advantages of each improved module and improves the overall detection ability of the model.

The results of the ablation experiment show that the *δ-SIOU* module can improve detection performance to a certain extent when used alone; however, there is no significant improvement with other individual modules in the improved basic model. The BIFPN structure alone affects the model’s detection performance. In contrast, the combination of the C3CA module and the BiFPN structure greatly enhances the detection performance of the aiming line. Analysis indicates that the introduction of the BiFPN structure improves the model’s fusion capabilities but increases its model complexity, thereby decreasing the model’s learning ability. Once embedded, the C3CA module improves the model’s ability to extract feature information about the target. When used individually or with other modules, *δ-SIOU* can improve detection performance, highlighting the benefits of the proposed detection performance improvement. The improved YOLOv5 model significantly enhances detection performance for the aiming line. The designed model maximizes the benefits of the improved modules, ultimately boosting overall detection abilities.

#### 4.3.2. Comparative Experiment

To evaluate the effectiveness of the aiming line detection model for the Gunner’s Primary Sight control system, we assessed classical models such as YOLOv3, YOLOv4, original YOLOv5, and YOLOv6 target detection models of the YOLO series. Additionally, we conducted a comparative experiment using the two-stage detection algorithm Faster R-CNN, as well as the one-stage detection algorithms SDD [34] and RetinaNet [35]. We conducted a comprehensive evaluation and analysis based on training time, reasoning time, and mean average precision (*mAP*). Table 5 shows a comparison between the training and inference time for different detection models on aiming line detection. The comparison results of *mAP* for different detection models on aiming line detection are illustrated in Figure 19.

Table 6 presents the training and inference speeds of various object detection models. The results indicate that while Faster R-CNN has a swift training speed, its inference speed is comparatively slower than the other models due to its network structure. YOLOv3 through YOLOv6 exhibit varying improvements in both training and inference speeds. However, the improvement from YOLOv6 to YOLOv5 is not significant. This study introduces an upgraded YOLOv5 model, which not only simplifies the original model but also ensures faster training and inference speeds, thereby making it better suited for industrial applications requiring real-time performance.

The present study proposes an improved YOLOv5 model that outperforms other models in terms of *mAP,* as shown in Figure 19. Specifically, compared to the original YOLOv5 model, the proposed model achieves a 2.1 percentage point increase, reaching an *mAP* of 96.7%. Additionally, the improved YOLOv5 model shows higher accuracy than Faster R-CNN, a two-stage detection algorithm. These findings suggest that the proposed model has superior detection performance, especially for aiming line detection in the gunner’s primary sight control system.

#### 4.3.3. The Detection Results of Aiming Line

This paper aims to verify the effectiveness and accuracy of the improved YOLOv5 gunner’s sight line detection method. The aiming line images in Gunner’s Primary Sight set were chosen for detection under varying conditions. Figure 20 shows the image detection results. The results revealed a precise and accurate detection of the aiming line, verifying Gunner’s Primary Sight detection model based on the YOLOv5 algorithm studied in this paper.

This paper aimed to further verify the effectiveness and accuracy of the proposed improved YOLOv5 model in measuring Gunner’s Primary Sight’s aiming line deviation. The study utilized the proposed method to calculate the image-stabilization deviation. The target image was detected using the improved YOLOv5 model, and then the position of the target’s border in the image was identified. The image stabilization deviation was calculated by determining the difference in position between the target borders in two frames. In order to simulate realistic tank driving conditions under a steady image state, the paper employed a mirror control test platform for experimental verification. Figure 21 and Figure 22 illustrate the variation curve of the image stabilization deviation.

As can be observed from the position change curve, the position change curve demonstrates a maximum error of approximately 5 pixels when using the method proposed in this paper, indicating high detection accuracy that enhances the precision of Gunner’s Primary Sight image stabilization.

## 5. Conclusions

In this paper, an image-stabilization error detection method for Gunner’s Primary Sight is presented in combination with improved. Improvements were implemented to enhance the YOLOv5 architecture. Firstly, the SPPF module was substituted by the SPPCSPC module, then the C3 module was augmented by embedding the CA attention resulting in the C3CA module, and the Neck network was optimized using the BIFPN structure. Further, a superior loss function, referred to as *δ-SIOU*, was introduced. Experimental results validate that the proposed algorithm not only delivers faster performance but also achieves higher detection accuracy in comparison to competing models. Furthermore, based on the calculated stabilization deviation, the measurement error of the algorithm is confirmed to be within five pixels, thus demonstrating its efficacy in detecting stabilizer deviation through targeting. It can provide a certain reference for the measurement of the aiming line of Gunner’s Primary Sight in the future, and it can also provide a certain reference for similar target detection.

In the future, we will further optimize the model structure to improve detection speed and reduce measurement errors in stabilization deviation. In addition, we will research model lightweight technology that allows detection models to be more easily deployed on mobile devices, such as portable detectors so that it can be fully used in the engineering field.

## Figures and Tables

**Figure 1 sensors-23-05039-f001:**
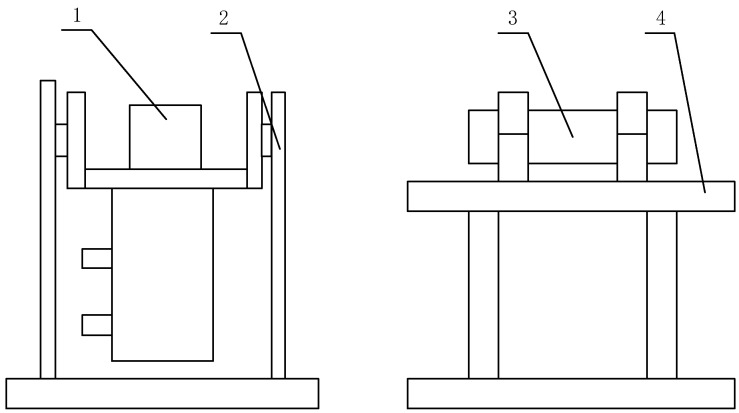
The schematic diagram of the collection device. Where 1 means the sight, 2 means the mirror control system bracket, 3 means the collimator, and 4 means the collimator installation bracket.

**Figure 2 sensors-23-05039-f002:**
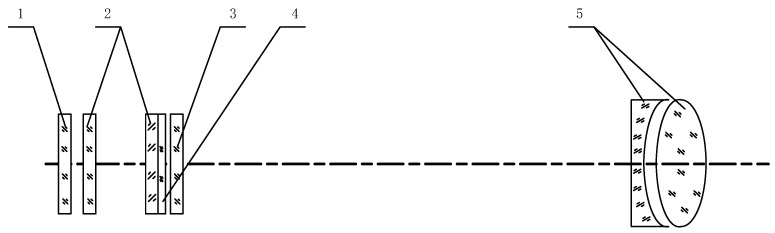
The schematic diagram of the collimator collection device. Where 1 is the frosted glass, 2 is the milky glass, 3 is the reticle, 4 is the filter, and 5 is the objective lens group.

**Figure 3 sensors-23-05039-f003:**
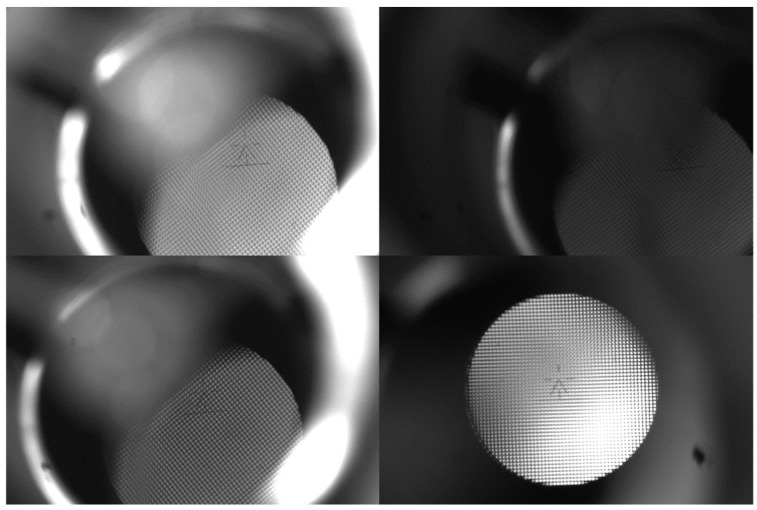
The aiming line image acquisition example.

**Figure 4 sensors-23-05039-f004:**
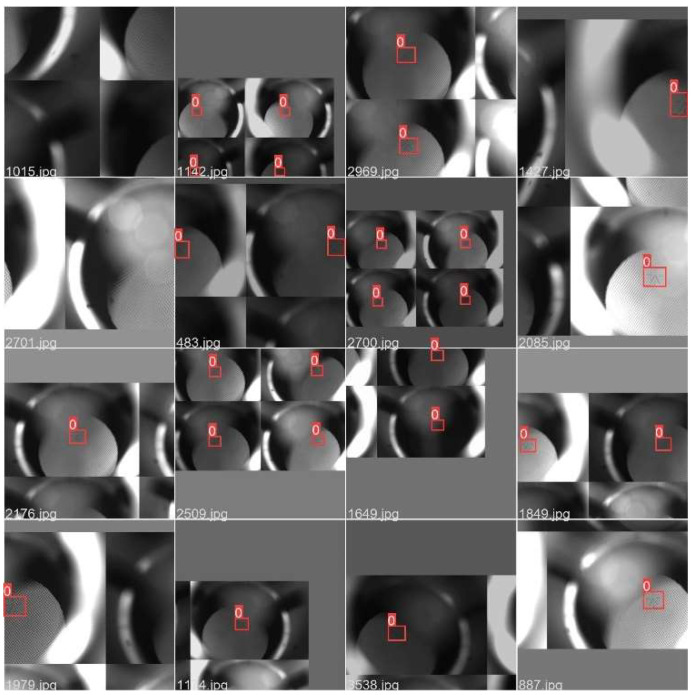
Data enhancement.

**Figure 5 sensors-23-05039-f005:**
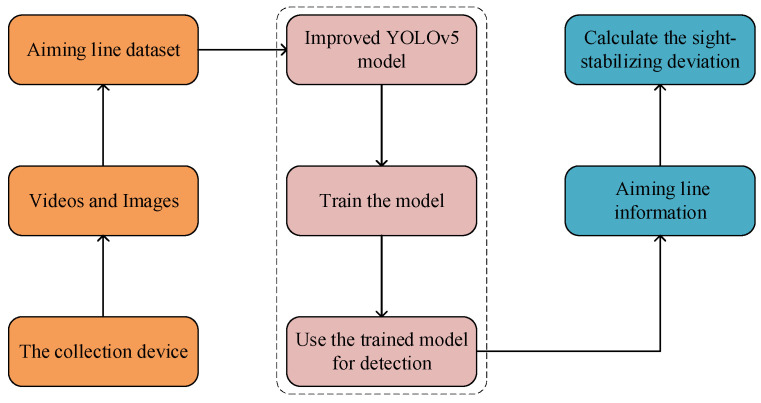
Diagram based on the modified YOLOv5s model to aiming line detection.

**Figure 6 sensors-23-05039-f006:**
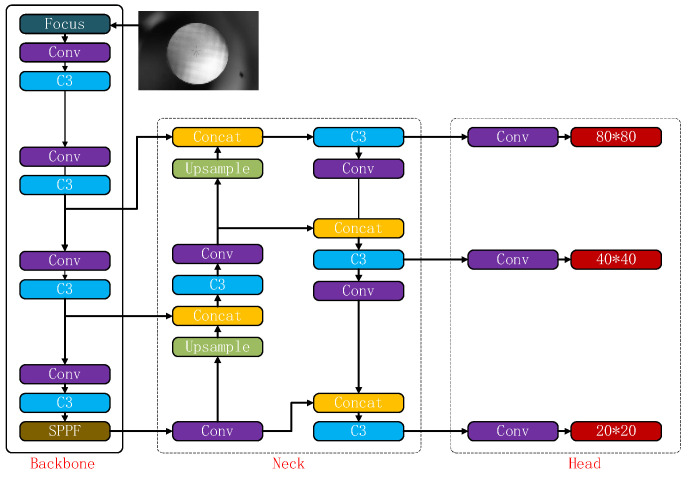
YOLOV5 network structure.

**Figure 7 sensors-23-05039-f007:**
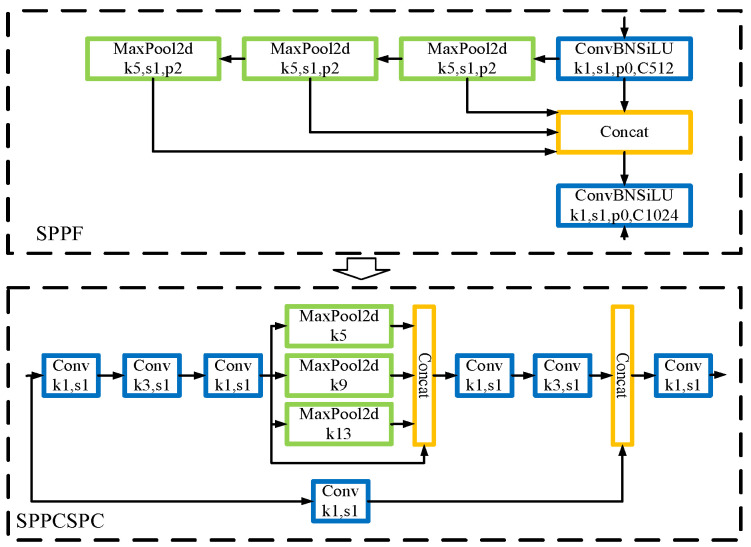
The schematic diagram of SPPF and SPPCSPC structure.

**Figure 8 sensors-23-05039-f008:**
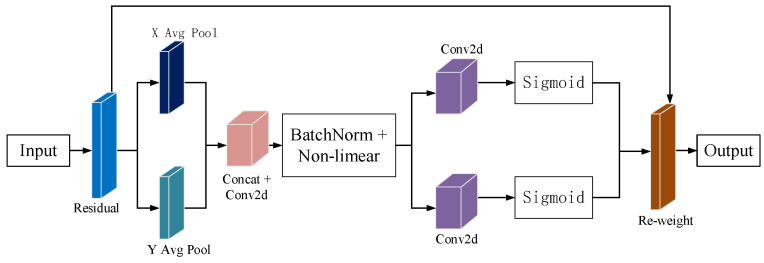
The structure of the CA attention module.

**Figure 9 sensors-23-05039-f009:**
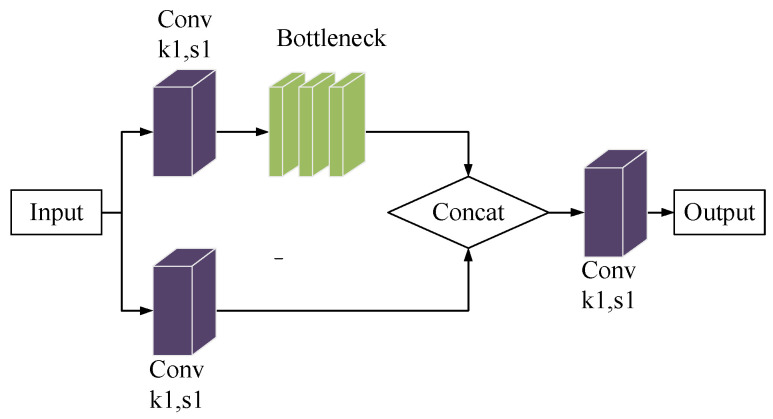
The structure of the C3 module.

**Figure 10 sensors-23-05039-f010:**
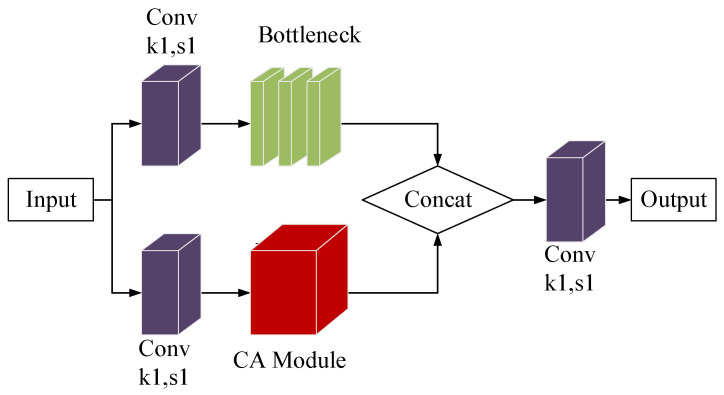
The structure of the C3CA module.

**Figure 11 sensors-23-05039-f011:**
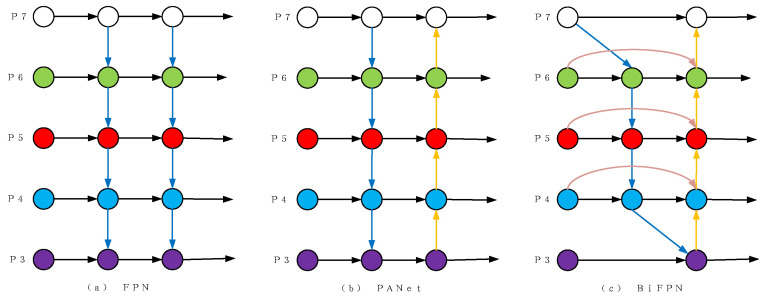
The schematic diagram of the feature fusion network.

**Figure 12 sensors-23-05039-f012:**
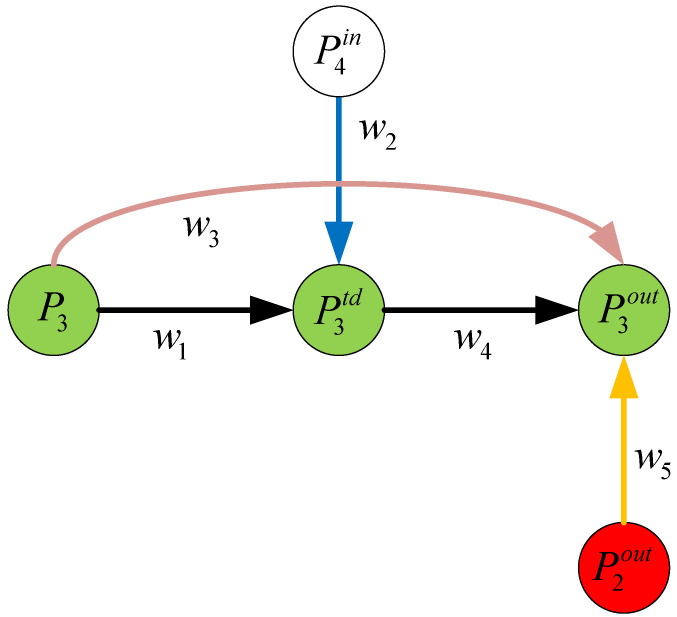
The schematic diagram of the other parameters and operations.

**Figure 13 sensors-23-05039-f013:**
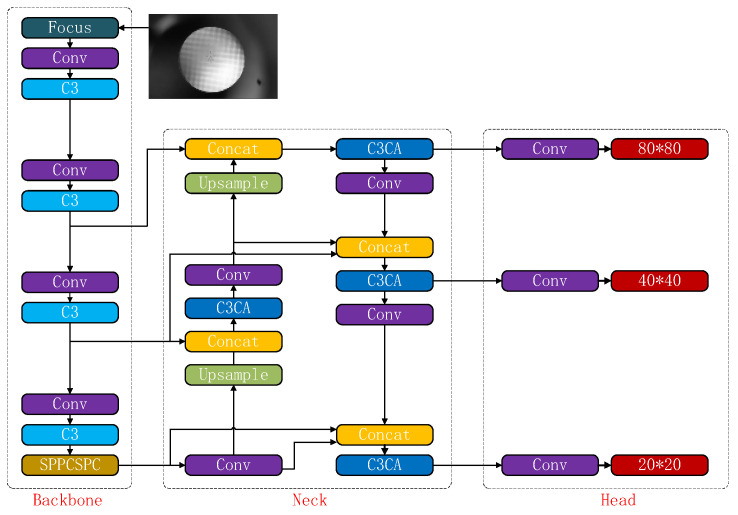
The structure of improved YOLOv5 network.

**Figure 14 sensors-23-05039-f014:**
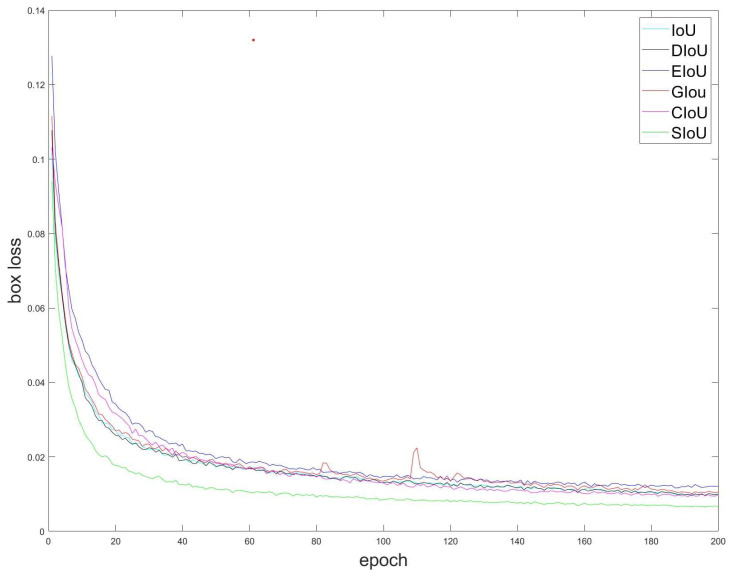
Different regression loss curves of *SIOU*.

**Figure 15 sensors-23-05039-f015:**
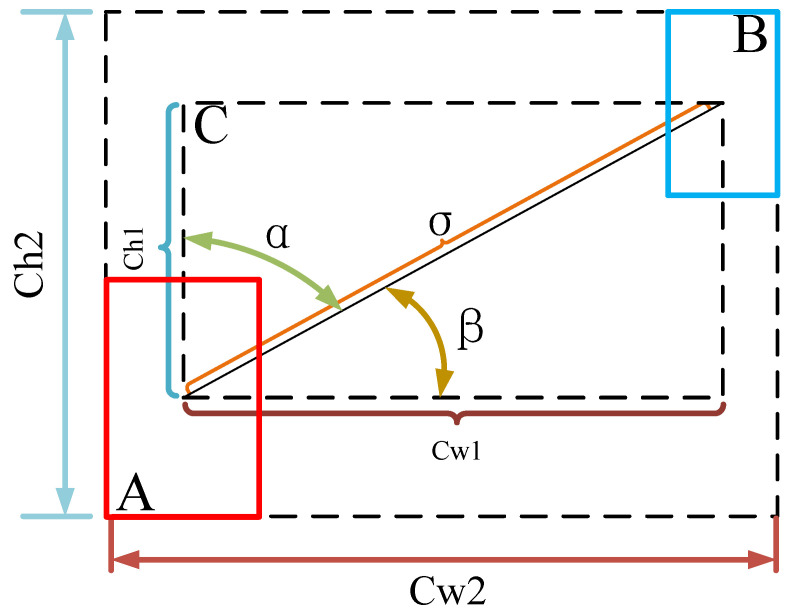
The schematic diagram of *SIOU*.

**Figure 16 sensors-23-05039-f016:**
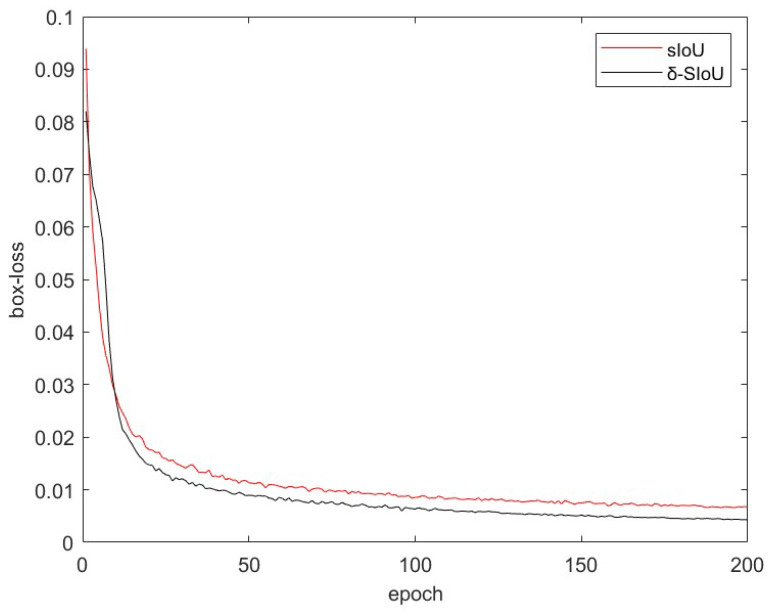
Comparison of regression loss curves of *SIoU* and *δ-SIOU*.

**Figure 17 sensors-23-05039-f017:**
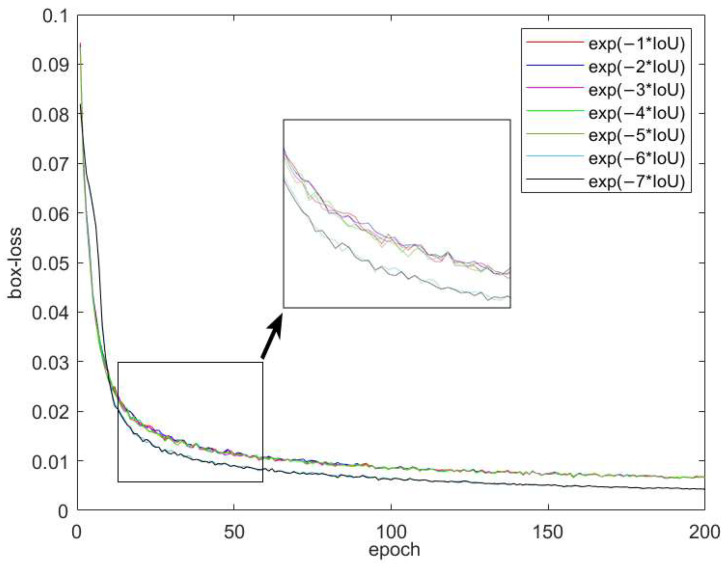
Comparison of loss curves of different coefficients on the *SIoU*.

**Figure 18 sensors-23-05039-f018:**
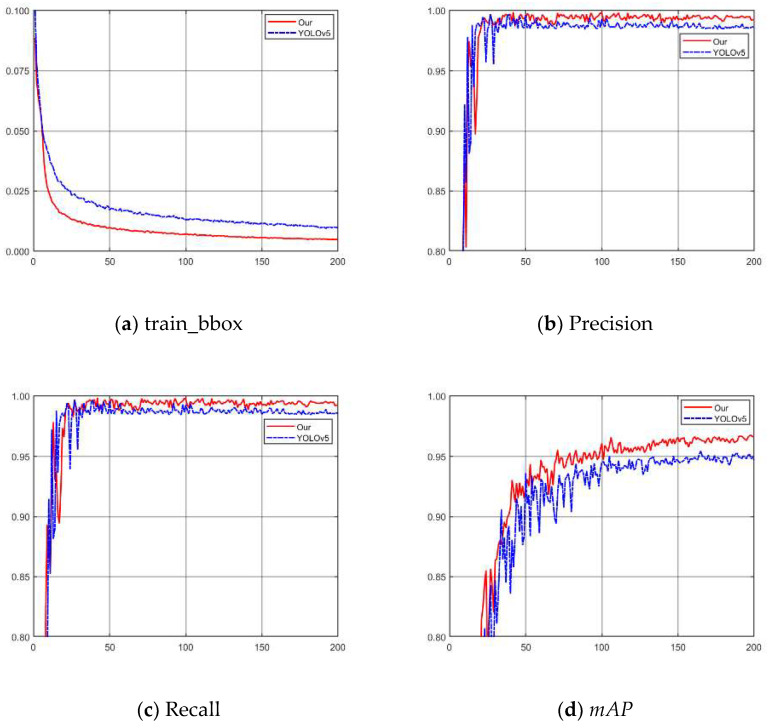
Comparison of training results with YOLOv5. (**a**) Correlation curve of train_bbox. (**b**) Correlation curve of Precision. (**c**) Correlation curve of Recall. (**d**) Correlation curve of *mAP*.

**Figure 19 sensors-23-05039-f019:**
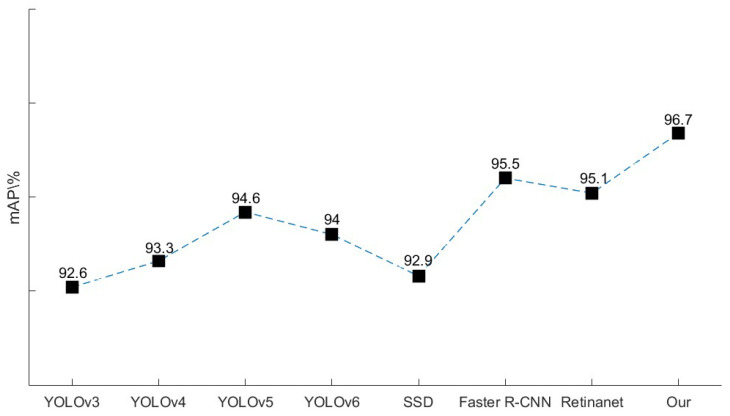
Comparison of average accuracy of different detection models.

**Figure 20 sensors-23-05039-f020:**
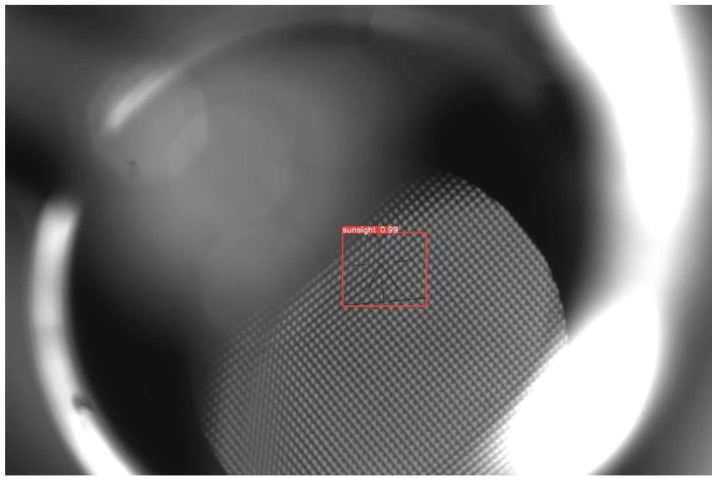
YOLOv5 gun leader’s sight line detection results.

**Figure 21 sensors-23-05039-f021:**
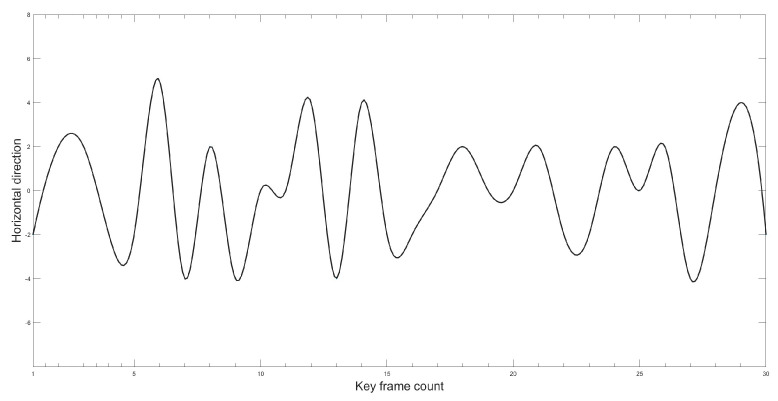
Error curve of the horizontal direction.

**Figure 22 sensors-23-05039-f022:**
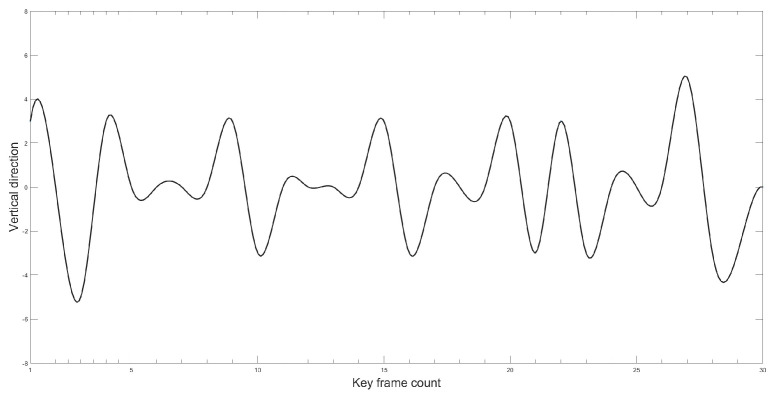
Error curve in the vertical direction.

**Table 1 sensors-23-05039-t001:** Data division.

Dataset	Training	Testing	Verification	Total
Line of sight Images	6040	1725	864	8629

**Table 2 sensors-23-05039-t002:** The comparative results of the space pyramid pool operation operation speed.

Name	Times/ms	Parameters
SPP	146.1487	7,225,885
SPPF	135.0896 (+11.0591)	7,235,389
SPPCSPC	120.2242 (+25.9245)	13,663,549

**Table 3 sensors-23-05039-t003:** Experiment configuration table.

Name	Model
CPU	AMD Ryzen 7 5800H, 3.20 GHz
GPU	NVIDIA GeForce RTX 3070 (8G)

**Table 4 sensors-23-05039-t004:** Experimental results of different regulatory factors.

The Value of the δ	*mAP*/%
$\delta =e^{\left(-6IoU\right)}$	94.8
$\delta =e^{\left(-7IoU\right)}$	94.6

**Table 5 sensors-23-05039-t005:** Ablation experiment.

δ − SIoU	BiFPN	C3CA	SPPCSPC	mAP/%
√				94.8
	√			93.5
		√		94.3
			√	94.4
√	√			94.7
√		√		94.6
√			√	94.8
	√	√		95.1
	√		√	94.6
		√	√	94.3
√	√	√		95.7
√	√		√	94.9
√		√	√	94.8
	√	√	√	95.3
√	√	√	√	96.7

**Table 6 sensors-23-05039-t006:** Training time and reasoning time of different detection models.

Model	Train Time (ms)	Inference Time (ms)
YOLOv3	167	24.7
YOLOv4	147	23.5
YOLOv5	203	22.6
YOLOv6	183	22.7
SSD	138	30.4
Faster R-CNN	152	31.2
Retinanet	186	24.1
Our	159	23.9

## Data Availability

Not applicable.

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
