# Peer review of "An Image Detection Method for Image Stabilization Deviation of the Tank Gunner’s Primary Sight"

_sensors, 2023, doi:10.3390/s23115039_

Round 1
Reviewer 1 Report
The following revisions are required.
1. In literature review, add 3 to 5 more relevant and latest techniques.
2. Add Comparison table at the end of section 2 and compare with at least 7 to 10 techniques with appropriate parameters.
3. Please make sure your paper has necessary language proof-reading.
4. The future scope is vague and appears too generic. Authors should revise the future scope and specify clearly the sorts of content.
5. The conclusion is weak and inconsistent with the evidence and arguments.
6. The methodology section needs to be stronger and provide the reason of improvement in Figure 18 and 19.
Please make sure your paper has necessary language proof-reading.
Author Response
Dear Reviewers:
We are very grateful to Reviewer for reviewing the paper so carefully. We have carefully considered the suggestion of Reviewer and make some changes.
1.In literature review, add 3 to 5 more relevant and latest techniques.
Response: We appreciate it very much for this good suggestion, and we have done it according to your ideas. We have inserted relevant reviews in lines 56-65 of the article.
2.Add Comparison table at the end of section 2 and compare with at least 7 to 10 techniques with appropriate parameters.
Response: We added relevant experiments in Table 5, bringing the total number of comparative experiments to 8, and made modifications to Figure 19.
3.Please make sure your paper has necessary language proof-reading.
Response: It is a great suggestion, as the reviewer pointed out, to have the language of the article proofread. We have rechecked the article and made corrections.
4.The future scope is vague and appears too generic. Authors should revise the future scope and specify clearly the sorts of content.
Response: Considering the Reviewer’s suggestion, We added our future prospects in the conclusion.
5.The conclusion is weak and inconsistent with the evidence and arguments.
Response: According to your suggestion, We have made modifications to the original conclusion.
6.The methodology section needs to be stronger and provide the reason of improvement in Figure 18 and 19.
Response: Thanks to the reviewer's feedback, we have modified the explanations of the relevant pictures.
Reviewer 2 Report
This manuscript proposes an improved one-stage image detection algorithm which detects the image stabilization of the tank gunner's primary sight, based on enhanced YOLOv5 sight-stabilizing deviation algorithm. To enhance the performance of YOLOv5's Backbone network, authors replace the SPPF module with the SPPCSPC module, which improves the model's features, as is the multi-scale target fusion ability, detection speed, and detection performance. Then, they propose a solution for neck network improvement by combining the BIFPN and C3CA modules. Finally, using the SIoU loss function with dynamic weights authors show that the proposed model has a high detection accuracy, as well as fast detection speed.
In my opinion, the manuscript is interesting, as it contains several new ideas in improving, optimizing and stabilizing the deviation of the primary sight of the tank gunner. Therefore, I think that it can be considered for acceptance, but with some extensive technical corrections, because there are several typos and ambiguities. I recommend the authors to read the manuscript again carefully and revise it thoroughly. Here are some of my suggestions:
1. Most abbreviations (IOLOv5, SIOU, C3CA, etc.) should be explained immediately when they appear, even in the abstract.
2. The sentence in lines 120-121 is not clear. Perhaps it should be connected with the caption on Figure 2.
3. Figure 3 showing the schematic diagramof the reticle is not necessary.
4. I do not see the verification set in Table 1.
5. Please, replace the heading of subsection 3.2. "Figures, Tables and Schematics" with a more inspiring title.
6. Please, delete the terms "This is a table. Tables should b" in Lines 190-191.
7. Line 285: Please, replace "rat" with "rate" and "It" with "it". Also, rewrite this whole sentence more "grammatically".
8. Line 327: Please replace "fferent" with "Different".
9. Please, rewrite Equation 6 more "mathematically".
Author Response
Dear Reviewers:
We are very grateful to Reviewer for reviewing the paper so carefully. We have carefully considered the suggestion of Reviewer and make some changes.
- Most abbreviations (IOLOv5, SIOU, C3CA, etc.) should be explained immediately when they appear, even in the abstract.
Response: Thanks to the reviewer's feedback, we have revised the abstract and related content according to their suggestions.
- The sentence in lines 120-121 is not clear. Perhaps it should be connected with the caption on Figure 2.
Response: We are very sorry for our incorrect writing and it is rectified at Line 120-121.
- Figure 3 showing the schematic diagramof the reticle is not necessary.
Response: We agree that Figure 3 is not necessary. The figure 3 has been removed.
- I do not see the verification set in Table 1.
Response: We are very sorry for our negligence of the verification set in Table 1. We have added content to the relevant section.
- Please, replace the heading of subsection 3.2. "Figures, Tables and Schematics" with a more inspiring title.
Response: We are very sorry for our incorrect writing and it is rectified at the heading of subsection 3.2.
- Please, delete the terms "This is a table. Tables should b" in Lines 190-191.
Response: Thank the reviewer for their careful reading of this article. We are sorry for the mistakes that happened. We have made the relevant corrections.
- Line 285: Please, replace "rat" with "rate" and "It" with "it". Also, rewrite this whole sentence more "grammatically".
Response: We have modified the grammatical errors in the original text according to your suggestions. In addition to that, we have also proofread the grammar of the original text.
- Line 327: Please replace "fferent" with "Different".
Response: We are very sorry for our incorrect writing and it is rectified at Line 327.
- Please, rewrite Equation 6 more "mathematically".
Response: It is really a good idea as Reviewer suggested, and we have changed it all to meet Reviewer’s thoughts.
Reviewer 3 Report
This study proposes an image detection method aimed at Gunner's Primary Sight control system of a specific tank that utilizes an enhanced YOLOv5 sight-stabilizing deviation algorithm. Initially, a dynamic weight factor is integrated into SIOU, creating δ-SIOU, which replaces CIoU as the loss function of YOLOv5.After that, the Spatial Pyramid Pool module of YOLOv5 was enhanced to improve the multiscale feature fusion ability of the model, thereby elevating the performance of the detection model. Lastly, the C3CA module was created by embedding the CA attention mechanism into the C3 module. The BiFPN network structure was also incorporated into the Neck network of YOLOv5 to improve the model's ability to learn target location information and image detection accuracy. Using data collected by a mirror control test platform, experimental results indicate an improvement in the detection accuracy of the model by 2.1%. The manuscript’s results are reproducible based on the details given in the methods section. However, all figures with charts could be bigger because they couldn’t be read. Also, I think the paper needs in conclusion to mention more about their future work. Moreover, what do they think about the new versions YOLOv7 or 8? Are more validated, more improved, more detection, more accurate and more speed.
Minor editing of English language required
Author Response
Dear Reviewers:
Firstly, thanks to the reviewer for their approval. We are very grateful to Reviewer for reviewing the paper so carefully. We have carefully considered the suggestion of Reviewer and make some changes.
Response: We have explained our future work at the end of the article. We have also rewritten the conclusion.
Reviewer 4 Report
Following are my comments and suggestions:
1. Title needs to be changed.
2. Organization of the article is required.
3. What is the improvement in YOLOv5 model? It is not clear from the article.
4. A Flowchart of the article is required.
5. Conclusion needs to be extended.
6. Typos in Eqs. 4 & 5. Use big bracket notation after Conv function.
7. There are some recent works on digital image processing. The authors can include in the reference list:
a) https://doi.org/10.1016/j.eswa.2021.115637
b) https://doi.org/10.1016/j.advengsoft.2022.103370
Author Response
Dear Reviewers:
Firstly, thanks to the reviewer for their approval. We are very grateful to Reviewer for reviewing the paper so carefully. We have carefully considered the suggestion of Reviewer and make some changes.
- Title needs to be changed.
Response: We are very sorry for our incorrect writing and it is changed at title of 3.2.
- Organization of the article is required.
Response: It is really a good idea as Reviewer suggested. We added a description of the article organization in lines 99-103.
- What is the improvement in YOLOv5 model? It is not clear from the article.
Response: We are very sorry for our negligence of the explanation. We added relevant explanations in lines 167-174 of the paper.
- A Flowchart of the article is required.
Response: It is really a great suggestion as Reviewer pointed out that add a flowchart. We have added the corresponding flowchart as suggested, as shown in Figure 5.
- Conclusion needs to be extended.
Response: We rewrote the conclusion and presented our future work.
- Typos in Eqs. 4 & 5. Use big bracket notation after Conv function.
Response: We are very sorry for our incorrect writing and it is rectified at Eqs. 4 & 5.
- There are some recent works on digital image processing. The authors can include in the reference list:
Response: Thanks to the literature recommended by the reviewer, we have agreed to insert it into this paper after reading it.